# Collagen Scaffolds Laden with Human Periodontal Ligament Fibroblasts Promote Periodontal Regeneration in SD Rat Model

**DOI:** 10.3390/polym15122649

**Published:** 2023-06-12

**Authors:** Yi-Tao Chang, Chuan-Ching Lai, Dan-Jae Lin

**Affiliations:** 1Graduate Institute of Clinical Medical Science, School of Medicine, China Medical University, Taichung 404, Taiwan; chang.petercet@gmail.com; 2School of Dentistry, College of Dentistry, China Medical University, Taichung 404, Taiwan; 3Department of Post-Baccalaureate Veterinary Medicine, Asia University, Taichung 413, Taiwan; 4Department of Physical Therapy, Asia University, Taichung 413, Taiwan; 5Department of Biomedical Engineering, College of Biomedical Engineering, China Medical University, Taichung 404, Taiwan

**Keywords:** collagen hydrogel, dental light-emitting diode, orientation of periodontal ligaments, periodontal regeneration, photo-cross-linking, riboflavin

## Abstract

Periodontitis, a chronic inflammatory disease caused by microbial communities carrying pathogens, leads to the loss of tooth-supporting tissues and is a significant contributor to tooth loss. This study aims to develop a novel injectable cell-laden hydrogel consisted of collagen (COL), riboflavin, and a dental light-emitting diode (LED) photo-cross-linking process for periodontal regeneration. Utilizing α-SMA and ALP immunofluorescence markers, we confirmed the differentiation of human periodontal ligament fibroblasts (HPLFs) into myofibroblasts and preosteoblasts within collagen scaffolds in vitro. Twenty-four rats with three-wall artificial periodontal defects were divided into four groups, Blank, COL_LED, COL_HPLF, and COL_HPLF_LED, and histomorphometrically assessed after 6 weeks. Notably, the COL_HPLF_LED group showed less relative epithelial downgrowth (*p* < 0.01 for Blank, *p* < 0.05 for COL_LED and COL_HPLF), and the relative residual bone defect was significantly reduced in the COL_HPLF_LED group compared to the Blank and the COL_LED group (*p* < 0.05). The results indicated that LED photo-cross-linking collagen scaffolds possess sufficient strength to withstand the forces of surgical process and biting, providing support for HPLF cells embedded within them. The secretion of cells is suggested to promote the repair of adjacent tissues, including well-oriented periodontal ligament and alveolar bone regeneration. The approach developed in this study demonstrates clinical feasibility and holds promise for achieving both functional and structural regeneration of periodontal defects.

## 1. Introduction

Periodontal diseases are widespread in both developed and developing countries, affecting approximately 20–50% of the global population. The high prevalence of periodontal disease in adolescents, adults, and older individuals is a significant public health concern [1,2,3]. According to a previous study in Taiwan, the occurrence of periodontitis has increased significantly from 11.5% in 1997 to 19.59% in 2013 (*p* < 0.0001). The average age of individuals with periodontitis has decreased from 54.46 ± 14.47 years old in 1997 to 45.51 ± 16.58 years old in 2013 [4]. Periodontitis can lead to the destruction of tooth-supporting tissues, such as alveolar bone, periodontal ligament (PDL), and root cementum. There are several approaches to treating periodontal disease, including guided tissue regeneration, application of enamel matrix derivatives, and various growth factors [5,6,7,8,9,10,11,12]. However, there are currently insufficient techniques to regenerate complete periodontal tissue, including alveolar bone, cementum, and well-oriented collagen fibers [13,14].

The prognosis of periodontitis in clinical dentistry is determined by the level of alveolar bone height and pocket depth. In general periodontal treatment, the result is limited to the healing of the long junctional epithelium, without the restoration of the PDL’s directional connection between the tooth and bone or true recovery of the alveolar bone height. However, the dental community seeks regeneration, which includes increasing alveolar bone height and rebuilding the directional connection of the PDL between tooth and bone. The regenerated structure can therefore act as a barrier to prevent bacteria or external forces from damaging the interior, while also allowing teeth to provide a stable biting force by connecting with the alveolar bone through the well-oriented PDL [15,16,17].

The research on stem cells has emerged as a promising field for tissue regeneration and the application of regenerative medicine [9,18]. The dental tissues have been found to contain adult stem cells that possess multipotent capabilities for self-renewal and differentiation into numerous types of cells [19,20,21,22]. The isolation of adult stem cells from human PDL tissues, referred to as periodontal ligament stem cells (PDLSCs), has opened up new opportunities for dental tissue engineering [13]. The use of stem cells in the right environment will allow the development of tissue regeneration with the ability to repair complex structures. The benefit of using pluripotent stem cells is also attributed to their secretion of growth factors, cytokines, and extracellular vesicles, including exosomes [23]. Periodontal ligament fibroblasts (HPLFs), despite their similar morphology to gingival fibroblasts (GFs), appear to exhibit their distinct functional activities in maintaining tissue integrity [24,25]. Human periodontal ligament cells (PDLCs) are considered as promising seed cells in periodontal tissue engineering. PDLCs show an osteoblastic phenotype upon osteogenic induction, which produces bone matrix and forms mineralized nodules [26,27,28,29,30,31]. A previous in vitro study also found that both Wnt3α and TGF-β1 promote myofibroblast differentiation from immature hPDLCs especially under forces [32]. The main component of PDL fibers is type I collagen [16]. Collagen gels that mimic the structure of natural extracellular matrices are considered as in vivo-linked cell culture methods and have been applied in PDLC-related research [26,27,28,29,30,31,33]. However, without cross-linking, the strength of the collagen gel is too low to meet clinical requirements, i.e., lacking sufficient strength to withstand the forces of surgical process and biting. The success of tissue-engineered bone regeneration is influenced by various factors that include the use of appropriate scaffolds/constructs for harvesting cells at the defect site, suitable cell type, adequate vascularization, signaling molecules for osteogenic differentiation, etc. Hence, a multidisciplinary approach will be required to develop newer graft materials that possess enhanced properties to obtain more desirable results [34,35]. Previous researchers used riboflavin, also known as vitamin B2, as a photoinitiator. Unlike other commonly used initiators, riboflavin has minimal cytotoxicity, making it a more biocompatible option. Additionally, the photo-cross-linking of collagen with riboflavin has been shown to improve mechanical properties and delay the degradation of collagen scaffolds [36]. Interestingly, scholars in the fields of ophthalmology and orthopedics have also used riboflavin and UVA as adhesives [37,38,39,40,41].

In clinical medicine, the dental LED device is commonly utilized for light-curing resin materials to repair dental hard tissues. In previous experiments, UV was used as the curing light source. During the photo-polymerization process, free radicals were produced, leading to cell death, with the length of illumination time inversely affecting the cell survival rate [38,39,40,41,42]. The use of a dental LED, which can be used near clinical operations, decreases the threshold for surgical technology requirements. A dental LED is a readily available resource that causes minimal damage to cells, making it a convenient option for future clinical use. Animal models that have been used for periodontal regeneration studies include dogs, rabbits, swine, and primates [9,18,43,44,45,46,47,48,49,50]. Each model has its advantages and disadvantages, depending on the specific research questions being addressed. In rat models, periodontal defects are created by surgically removing part of the periodontal ligament and alveolar bone around the tooth and using them to study periodontal regeneration and potential therapies [47,48].

This study aimed to develop a novel technique for regenerating periodontal tissue with well-oriented collagen fibers. A cell-laden collagen gel containing riboflavin was photo-polymerized with a dental LED light, and the scaffold mechanics and cell viability were tested in vitro for obtaining the optimal cell-seeding and photo-cross-linking condition. An animal study using Sprague Dawley rats was conducted to verify periodontal tissue regeneration under different injection conditions compared to a blank group. The hypothesis of this study is that HPLF cells housed in the injected collagen can be photo-cross-linked in situ and benefit the regeneration of periodontal defects. The null hypothesis of this study is that the injectable collagen hydrogel with appropriated mechanical strength combining cell therapy will not affect the PDL orientation, epithelial downgrowth, and bone regeneration of the artificial periodontal defects.

## 2. Materials and Methods

### 2.1. Human Periodontal Ligament Fibroblasts

Human periodontal ligament fibroblasts (HPLFs) (ScienCell, Cat. No. 2630) were cultured in fibroblast medium (ScienCell, Cat. No. 2301), which consisted of 100 mL of basal medium, 10 mL of fetal bovine serum (FBS, Cat. No. 0010), 5 mL of fibroblast growth supplement (FGS, Cat. No. 2352), and 5 mL of antibiotic solution (P/S, Cat. No. 0503). After embedding the cells within the collagen scaffold, we cultured them further in osteogenic differentiation medium (ScienCell, Cat. No. 7531) [30].

### 2.2. Photo-Cross-linking of Collagen and Cell Encapsulation

We prepared collagen hydrogels (Sigma, bovine type I collagen, 0.27% *w*/*v*) by mixing 1 part of 10 × PBS with 9 parts of collagen solution (0.3% *w*/*v* in 0.01 N acetic acid) and adding 0.01% riboflavin (RF) solution. The resulting mixture was then neutralized with 0.1 M NaOH [38,44]. We gently mixed the collagen gel with HPLF cells and transferred the mixture to a 48-well plate (concentration: 1 × 10^6^ cells/300 uL construct). To achieve light cross-linking and coagulation, we used a blue LED with an intensity of 2000 mW/cm^2^ and a wavelength of 470 nm for 3, 9, and 21 s. The control group (0 s) was included without light exposure. Finally, we added fibroblast medium to the cell-laden collagen hydrogel and cultured it in a humidified incubator (37 °C, 5% CO_2_).

### 2.3. Strain Testing of Collagen Scaffold

A strain test (% strain) of the collagen structure was performed using a Handy HF series electronic tensile force gauge (JISC, ALGOL, Handy Force Gauge HF series) equipped with a 10 N loading cell to measure the strain (% strain) of the collagen structure. The cell–collagen mixtures (containing 0.01% RF) were cross-linked with a blue LED for 0, 3, 9, and 21 s, and applied strains of 0.5%, 1%, 1.5%, and 2%, respectively.

### 2.4. Cell Viability Test of HPLF Collagen Scaffolds

To determine whether the LED light affected cell viability, we used the LIVE/DEAD viability assay kit (Invitrogen, Cat. No. L3224) to test HPLF cell viability [38]. It provides two molecular probes with green-fluorescent calcein-AM that indicates esterase activity in live cells, and red-fluorescent ethidium homodimer-1 that indicates the loss of plasma membrane integrity in dead cells. Collagen hydrogels containing riboflavin (RF) were prepared and illuminated with LED as described above, then the HPLC cell viability (live cell %) was counted as live cells among all live and dead cells after culture for 24 h.

### 2.5. Cell Differentiation

Alpha-SMA (alpha smooth muscle actin, α-SMA) and ALP (alkaline phosphatase) are markers for myofibroblasts and pre-osteoblasts, respectively [51]. After 1, 7, and 14 d, cellular differentiation of HPLC cells in collagen scaffolds was explored by α-SMA and ALP immunofluorescent staining. Briefly, HPLF-laden collagen scaffolds were fixed with 4% paraformaldehyde in 0.1 M PB, pH 7.4, for 1 min and then blocked with 5% BSA for 30 min at room temperature. Samples were incubated with Recombinant Alexa Fluor^®^ 647 anti-α-SMA antibody (Abcam, Cat. No. ab196919, 1:100) or anti-ALP antibody (Abcam, Cat. No. ab126820, 1:100) in 1% BSA in PBS in a humidified chamber for 1 h at room temperature in the dark. Thereafter, the samples were incubated with a goat Anti-Mouse IgG-conjugated Alexa Fluor^®^ 488 (Abcam, Cat. No. ab150113) in 1% BSA in PBS for 1 h at room temperature in the dark. Finally, mounting medium with DAPI (Abcam, Cat. No. ab104139) was applied directly on top of the specimen and covered with coverslips. The expression of α-SMA and ALP was observed under the Dragonfly Confocal Microscopy System (Oxford, Andor) and conducted using an image analysis software (Image J, v. 1.53 t) [52].

### 2.6. Animals and Surgical Protocol

Twenty-four Sprague Dawley rats were housed in the Laboratory Animal Center of China Medical University in a temperature-controlled room (25 °C) under a 12 h/12 h light–dark cycle with free access to food and water. Animal handling was in accordance with a protocol approved by the Institutional Animal Care and Use Committee of China Medical University (CMUIACUC-2020-125). All efforts were made to minimize animal suffering and reduce the number of animals used. During the experimental periods, the animals were healthy and adapted to the treatment without stress. The animals were anesthetized by the intraperitoneal injection of ketamine hydrochloride (30 mg/kg) and xylazine hydrochloride (10 mg/kg), followed by the creation of bilateral intrabony three-wall defects. An incision was made to the maxillary first molar along the alveolar ridge. We flushed the incision with saline containing 0.1% adrenalin in subsequent steps to stop bleeding and remove debris. The flap was raised to expose the root surface and surrounding alveolar bone. A 1.4 mm round dental tungsten carbide drill was used to create defects, and a clinical periodontal probe was used to monitor the size and contour of the defect (2 × 1.4 × 1.4 mm^3^), as shown in Figure 1. Immediately after hemostasis with sterile gauze, the experiment defect was injected with collagen hydrogel by experimental pipet. The collagen hydrogel was prepared as described in Section 2.2.

The 24 animals were randomly divided into 4 experimental groups, as Blank, COL_LED, COL_HPLF, and COL_HPLF_LED. The Blank group did not insert any materials in the defect. The defect of the COL_LED group was injected with RF-containing collagen gel without any cell and then cross-linked by an LED for 3 s. The COL_HPLF group was injected with mixture of RF-containing collagen gel and HPLF cells. The COL_HPLF_LED group was injected with an LED cross-linking collagen mixture of RF-containing collagen gel and HPLF cells and then cross-linked by an LED for 3 s. After placement of the scaffold, the flap was repositioned and closed with absorbable sutures (Vicryl 5-0, Ethicon Products, Amersfoort, The Netherlands).

### 2.7. Histomorphometrical Analysis

Six weeks after surgery, the rats were deeply anesthetized and then sacrificed. Complete maxillae were retrieved and excess tissue was removed. After fixing with 4% paraformaldehyde in 0.1 M PB, pH 7.4, overnight, the samples were transferred to a buffered 50% formic acid decalcifying solution (Morse’s decalcification) for 45 d, and the solution was replaced three times per week. After demineralization, the samples were embedded in paraffin and sectioned into 4.0 μm thick cross-sections. The sections were stained with hematoxylin-eosin (HE) or Masson’s Trichrome (MT) for histomorphometric evaluation. The histological images were captured using an Olympus VS200 Research Slide Scanner and were analyzed with image analysis software (Image J, v. 1.53t).

The orientation of the PDL fibers was obtained by the following steps: First, the tissue slice image was converted from an RGB color image to an 8-bit grayscale image. Second, the observation area of the postoperative periodontal site was circled. Third, the ImageJ software was opened, and “Directionality” was selected in the “Analysis” functional catalog. Finally, the result was expressed as a direction distribution histogram.

To conduct the histomorphometrical analysis, we established a set of baseline and measurement targets for quantitative analysis, taking into consideration the histological structure of both teeth and alveolar bone as shown in Figure 2.

Tooth Length/Tooth Width (TL/TW): the length from the crown cusp to the apex/the crown width of the cemento-enamel junction (CEJ) levelLine of the adjacent tooth CEJLine of the adjacent tooth alveolar bone levelHeight of the epithelial downgrowth: the bottom of the periodontal pocket to the CEJHeight of the residual bone defect: the bottom of the bony defect to the bone levelAngle of the epithelium–tooth axis: the angle of the epithelium to the TL

Further minute adjustments were conducted for the “epithelial downgrowth” and “residual bone defect” to eliminate the possible bias from the histological process. The “relative epithelial downgrowth” and “relative residual bone defect” were measured as a ratio to tooth length (TL) or tooth width (TW) to ensure that the data and conclusions were more accurate and reliable.

### 2.8. Data Analysis

The data are presented as mean ± S.E.M. To detect the differences among groups, data were analyzed using SPSS 29.0 (IBM-SPSS) and one-way ANOVA with Fisher’s LSD or Tukey’s HSD post hoc comparison was applied. Significance levels were set at: * *p* < 0.05, ** *p* < 0.01, *** *p* < 0.001.

## 3. Results

### 3.1. Scaffold Characterization and Cell Viability

After the HPLF cells were combined with the collagen hydrogel, the mixtures were cross-linked with a blue LED for 0, 3, 9, and 21 s, and the strain rate of the collagen structure was measured at various time groups (Figure 3a). When a 0.5% strain was applied, the force measurements of 3 s (0.034 ± 0.003), 9 s (0.024 ± 0.008), and 21 s (0.028 ± 0.010) were higher than that of the 0 s group (0.013 ± 0.003). As a 1% strain was applied, the force measurements of 3 s (0.063 ± 0.050), 9 s (0.046 ± 0.009), and 21 s (0.053 ± 0.013) were higher than that of the 0 s group (0.019 ± 0.006). As a 1.5% strain was applied, the force measurements of 3 s (0.075 ± 0.053), 9 s (0.065 ± 0.010), and 21 s (0.076 ± 0.017) were higher than that of the 0 s group (0.028 ± 0.008). As a 2% strain was applied, the force measurements of 3 s (0.061 ± 0.041), 9 s (0.078 ± 0.008), and 21 s (0.092 ± 0.010) were higher than that of the 0 s group (0.019 ± 0.006). All the 3, 9, and 21 s groups had higher stiffness than the 0 s group (*** *p* < 0.001), and there were no differences among these LED-cured groups.

In order to confirm the condition of the cells after the dental LED illumination, the collagen cell scaffolds were cultured for 2 days, and the cell viability was detected by Live-Dead. The cell viability of 3 s (59.75 ± 0.81) under the LED light was higher than that of the 9 s (46.12 ± 1.19) and 21 s (45.12 ± 2.25) groups (** *p* < 0.01) (Figure 3b–e).

Considering the cell activity and the toughness of the collagen structure, LED light for 3 s was selected as the photocuring condition of the collagen scaffold in the following experiments.

### 3.2. Cell Differentiation

To understand the differentiation of HPLF cells, we selected α-SMA and ALP as markers for myofibroblasts and pre-osteoblasts, respectively. As previously described, the HPLF cells embedded in the collagen scaffold were cultured for 1, 7, and 14 days in osteogenic differentiation medium (Figure 4a). The expression of α-SMA, ALP, and DAPI was measured using immunofluorescence (Figure 4b).

DAPI expression was significantly lower on day 7 (74.6 ± 4.43%, * *p* < 0.05) compared to day 1 (100.0 ± 7.61%) but returned close to the baseline value on day 14 (99.6 ± 8.47%). The α-SMA/DAPI unit cell expression was significantly higher on day 7 (135.0 ± 6.60%, ** *p* < 0.01) compared to day 1 (100.0 ± 5.68%) but returned close to the baseline value on day 14 (106.7 ± 8.79%). Similarly, the ALP/DAPI unit cell expression was significantly higher on day 7 (147.6 ± 7.18%, ** *p* < 0.01) compared to day 1 (100.0 ± 6.71%) but returned close to the baseline value on day 14 (116.6 ± 11.55%). These results demonstrated that HPLF cells in osteogenic medium differentiate into myofibroblasts and preosteoblasts within a few days.

### 3.3. The Orientation of New Periodontal Ligament and Alveolar Bone

The periodontal ligament plays a physical and physiological role, of which orientation is an important function. After creating periodontal defects in SD rats, PDLFs and/or collagen gel were implanted into defects. Rats were sacrificed for histology and histomorphometrical analysis.

The Blank group showed that cells and fibers were evenly distributed, and the newly formed periodontal ligament was integrated from multiple angles, partly toward the tooth root. In the COL_LED group, the distribution of cells and fibers was disordered, and the newly formed periodontal ligaments were multi-directional and had no obvious connection with the tooth root. The COL_HPLF osteoblast clusters were evident with bone formation, but new periodontal ligaments were not obvious. The COL_HPLF_LED group showed osteoblast accumulation near the new bone, and the newly formed periodontal ligaments directionally connected the bone and the root surface (Figure 5). This finding indicated that periodontal ligaments were restored and functioned well in the COL_HPLF_LED group. The direction distribution histograms of the selected area beneath the regenerated PDL in typical histological pictures are shown in Figure 5c,f,i,l. The COL_HPLF_LED group demonstrated a more concentrated direction distribution compared to other groups, which indicated the COL_HPLF_LED group exhibited well-oriented PDL fibers in the 6-week periodontal regeneration.

### 3.4. Histomorphometry

Epithelial downgrowth occurs in response to inflammation or injury, and the least epithelial downgrowth may predict better postoperative recovery. All the data are shown as a percentage compared to the Blank group. The downgrowth of epithelial cells was significantly lower in the COL_HPLF_LED group (34.4 ± 9.94), compared to the COL_LED group (88.8 ± 9.57) (# *p* < 0.05) (Figure 6a). In order to avoid differences caused by individuals, we adjusted the data according to the tooth width and tooth length. Related to the tooth width, the downgrowth of the epithelium of the COL_HPLF_LED group (35.8 ± 10.75) was significantly lower than in the COL_LED group (95.4 ± 11.90) (# *p* < 0.05). Related to the tooth length, similarly, the downgrowth of the epithelium of the COL_HPLF_LED group (36.6 ± 11.08) was significantly lower than in the COL_LED group (107.1 ± 11.54) (## *p* < 0.01) and the COL_HLPF group (83.5 ± 12.12) (# *p* < 0.05) (Figure 6b). These data suggested that HPLF cells implanted with collagen scaffolds would prevent the downgrowth of the epithelium.

Restoration of defective bone indicated the regeneration capability of alveolar bone. We measured the residual bone defect under histological analysis. These data are also shown as a percentage compared to the Blank group. As shown in Figure 6c, the residual bone defect was significantly lower in the COL_HPLF_LED group (31.6 ± 7.29) compared to the COL_LED group (91.4 ± 22.94) (# *p* < 0.05). As previously described, to avoid differences caused by individuals, we also adjusted the data according to the tooth width and tooth length (Figure 6d). Related to the tooth width, the residual bone defect of the COL_HPLF_LED group (32.7 ± 7.93) was significantly lower than in the COL_LED group (98.8 ± 25.81) (# *p* < 0.05). Related to the tooth length, the residual bone defect of the COL_HPLF_LED group (34.8 ± 9.11) was significantly lower than in the COL_LED group (108.3 ± 26.75) (# *p* < 0.05). HPLF cells implanted with collagen scaffolds could promote the regeneration of alveolar bone.

The epithelium–axis angle is an indicator of epithelium and connective tissue repair, with small angles indicating increased soft tissue attachment to the root surface. The angle of the epithelium to tooth axis was significantly lower in the COL_HPLF_LED group (69.7 ± 8.40) compared to the COL_LED group (121.2 ± 3.50) (Figure 6e, # *p* < 0.05). This finding suggested that HPLFs can promote epithelial and connective tissue repair.

## 4. Discussion

Scaffold materials have been widely used in dental clinics. However, the interplay between transplanting cells and scaffold materials has not been carefully characterized. The combinational effects of collagen hydrogels and PDLCs, for example, are still not clear. In this study, we compared the role of scaffold materials and HPLF cells in periodontal structural repair. We observed enhanced regeneration following the dental LED hardening of collagen gels and PDLCs. The HPLF cells housed in the injected collagen can be photo-cross-linked in situ and benefit the regeneration of periodontal defects. In the COL_HPLF_LED group, epithelial downgrowth and residual bone defects were significantly reduced. In addition, the periodontal ligament in the COL_HPLF_LED group showed directional regeneration and connected the new bone and the root surface. Together, our results characterized the changes in bone and epithelium repair that may account for periodontal ligament orientation.

### 4.1. Orientation of Periodontal Ligament

The periodontal ligament is the connective tissue embedded between the cementum and the inner wall of the alveolar socket and has a specific orientation relative to the tooth root [14,15]. Destruction of the PDL and alveolar bone disrupts the balance between periodontal tissue and occlusal forces. Despite chemical and biological approaches, periodontal regeneration remains a challenge, and there are currently no techniques that can fully regenerate all components of periodontal tissue, including alveolar bone, cementum, and well-oriented collagen fibers [15]. A well-oriented periodontal ligament acts as a barrier between bone and epithelium to prevent bacterial invasion. In the present study, each group showed restoration of bone, epithelium, and connective tissue. However, complete periodontal regeneration of newly formed bone and cementum connected with well-oriented collagen fibers was observed only in the COL_HPLF_LED group. It is expected that HPLF cells grow well in the initial couple of days through the scaffold and promote the stable secretion of cytokines. They attract rat pluripotential cells near the collagen gel injection, result in formation of new bone, and reduce downgrowth of epithelial cells. We believe that the injected HPLFs play an important role in the initial stage, and after secreting many important messages, they die as the collagen degrades gradually. The detailed mechanism remains for further exploration. Akita et al. showed that the internal orientation of collagen fibers in the connective tissues influences the mechanical properties of the tissues and suggested that, in general, the collagenous bundles could best resist axially directed forces. The arrangement of the majority of the periodontal ligament collagen fibers are in horizontal and oblique directions and hence may be adapted to resist axial forces [16].

### 4.2. Epithelial Downgrowth and Residual Bone Defect

The movement of the junctional epithelium (JE) is a key factor in the advancement of periodontitis. The shift of the JE toward the apex, along with the deterioration of the crestal bone and periodontal ligament, is the characteristic manifestation of periodontal disease progression and attachment loss. Epithelial downgrowth refers to the migration of the outermost layer of oral epithelial cells into the connective tissue surrounding the tooth. This process occurs in response to inflammation or injury and leads to the formation of periodontal pockets, which can be sites of bacterial growth and periodontal disease. The presence of epithelial downgrowth complicates periodontal treatment and may require surgical intervention for removal.

A smaller residual bone defect indicated better postoperative bone regeneration capability. COL_HPLF_LED group had the smallest residual bone defect and therefore had the best postoperative recovery. In contrast, COL_LED had a larger residual bone defect, possibly because of the lack of cells in its scaffold, implying that there were no signal factors to attract osteoblasts to the surgical site. However, HPLFs can release signal factors and effectively attract osteoblasts, thereby promoting bone repair and regeneration [19,21,53]. Instead, it is thought that collagen can act as an epithelial barrier like a guided tissue regeneration principle to reduce downgrowth of the epithelium and preserve space for bone growth. However, the results of the COL_LED group did not show a significant effect in this study. This may be due to the collagen degradation time, which was insufficient to counteract the downgrowth of the epithelium and maintain space for bone growth. HPLFs reduce the epithelium–tooth axis angle by increasing the connection of the epithelial connective tissue to the root surface through oblique or horizontal fibers. This finding suggested that HPLFs restrict tissue growth in a horizontal manner that promotes periodontal regeneration, unlike vertically oriented fibers. In addition, HPLFs may compete with gingival mesenchymal stem cells for healing space and reduce epithelial downgrowth.

The cell–collagen hydrogel was injected into the surgical defect area using a laboratory pipette and was sutured after being cured with a dental LED light, which is in line with the clinical operation protocol technique. The COL_LED group formed a barrier such as the clinical guided tissue regeneration (GTR), preventing epithelial downgrowth and providing space for bone growth. However, without HPLF cell regulation, there was no significant difference compared to the Blank group. Although the COL_HPLF group had HPLF cell regulation, the strength of collagen was significantly insufficient without dental LED light cross-linking. HPLF cells significantly aggregated at the bottom of the surgical defect area and failed to spread over the entire surgical defect area, so there was no significant difference compared to the Blank group. The COL_HPLF_LED group had HPLF cell regulation and dental LED light cross-linking, resulting in significantly improved collagen toughness. HPLF cells were significantly dispersed in the surgical defect area, making bone residual defects or the degree of epithelial migration the lowest. The periodontal tissue regeneration was optimal and the results showed a statistically significant difference.

### 4.3. Evaluation of Periodontal Regeneration

The progress of periodontal regeneration can be evaluated by various methods, such as radiographic imaging and histological analysis, to measure bone height and attachment level around the tooth [9,18,43,44,48]. When assessing periodontal health, a thorough understanding of bone height is necessary for maintaining the health and integrity of the periodontal tissues [18]. It provides valuable information about the support and destruction of periodontal tissue and aids in diagnosing periodontal disease and developing appropriate treatment plans. In addition, it helps to monitor the progression of the disease over time and assess the effectiveness of treatment.

The cemento-enamel junction (CEJ) is a critical reference point for evaluating periodontal health. It serves as a fixed and static landmark that can be used to measure clinical attachment level and probe pocket depth, providing valuable information about the extent of periodontal destruction. Additionally, the CEJ can be used to assess alveolar bone destruction by measuring the distance between the CEJ and bone crest. By utilizing the CEJ as a reference point, dental professionals can accurately evaluate the health of the periodontal tissues and develop effective treatment plans to restore and preserve a patient’s periodontal health. The tooth root length (TL) and tooth crown width (TW) in the experiment were used for angle difference correction during tooth specimen slicing.

### 4.4. Human Periodontal Ligament Fibroblasts

Because of the relatively easy availability of cell sources, dental stem cells are currently used in the fields of neurology and orthopedics. Stem cells extracted from teeth can be differentiated into cells of different tissues such as osteoblasts, chondrocytes, and neurons, and are used in cell therapy and tissue engineering research. Moreover, since dental stem cells do not damage tooth tissue during processing, the harm and pain to patients are reduced, and it is a relatively safe and effective treatment method [54,55,56,57,58]. Human periodontal ligament fibroblasts (HPLFs) are in the periodontal ligament and used for periodontal regeneration because of their potential in bone formation. HPLFs imitate cells of the osteoblast lineage and express multiple osteogenic markers such as minerals, collagen, and alkaline phosphatase [31]. HPLFs are known to produce osteoblast-associated extracellular matrix proteins and display higher alkaline phosphatase activity compared with gingival fibroblasts [24,25]. HPLFs expressed lower immunogenicity and can be used in autologous, allogeneic, and xenogenic transplantation [55,59].

Expression of α-SMA and ALP represented markers for the presence of myofibroblasts and pre-osteoblasts, respectively [51]. In this study, the expression of α-SMA and ALP peaked at day 7 and then declined. Likewise, previous studies showed that the expression of ALP in PDLSCs increased at day 6, peaked at day 12, and then decreased at day 15 [60]. It had been established that as osteoblasts progress in differentiation, their capacity for cell division decreases, suggesting an inverse relationship between the proliferation rate and differentiation degree of osteoblasts [61,62]. This was consistent with the results of this experiment.

### 4.5. Animal Model

Periodontal disease is more common in adults. We used 12-week-old SD rats, which can closely mimic the clinical situation when considering the rats as adults. Because of collagen degradation times ranging from 3 to 23 days [15], we sacrificed the rats at 18 weeks old in this study. In previous studies, the periodontal ligament had a higher collagen turnover rate than the skin and oral mucosa, as measured by the uptake of tritiated proline and glycine in the periodontal ligament. Therefore, we opted for a longer recovery period to observe the effects of the treatment more accurately. In addition, we chose male rats to avoid the potential hormonal influences of pregnancy or the estrous cycle on the experimental results. This animal model is a surgical defect model, not a disease-induced defect model, so there may be differences in localized tissue phenotype. Disease-induced defect models can cause related diseases because of bacterial infection and inflammation that are difficult to control, and there are individual differences in disease resistance that can easily cause differences. Surgical defect models are commonly used experimental models because the size of the experimental area is relatively easy to control [48,63,64]

### 4.6. Scaffold Material and LED Light Source

In the comparison between the COL_HPLF and COL_HPLF_LED groups, the tensile strength of the collagen scaffold was increased by the LED exposure. This result suggested that the hardened collagen scaffold helps HPLFs maintain relative positioning and release regulatory factors, thereby enhancing the effectiveness of periodontal regeneration [65].

Hardening of the scaffold material requires the action of a photoinitiator and a specific light source. In the dental industry, a variety of photoinitiators are available, including hydrogen peroxide (H_2_O_2_), titanium dioxide, sodium dodecyl sulfate, and glutaraldehyde, among others. In this experiment, riboflavin (also known as Vitamin B2) was selected as the photoinitiator, which has minimal cytotoxicity and better biocompatibility compared to conventionally used photoinitiators. Additionally, the photo-cross-linking of collagen and riboflavin improves the mechanical properties and delays the degradation of collagen scaffolds. Interestingly, scholars in the fields of ophthalmology and orthopedics have also used riboflavin and UVA as adhesives [37,38,39,40,41].

In dental clinical medicine, LED devices are often used in the restoration of hard tissue with light-cured resin materials. In previous experiments, ultraviolet light was used as the curing light source, but free radicals were generated during photo-polymerization. This will negatively affect cell survival and even lead to cell death. The use of dental LEDs that can be used near clinical procedures lowers the threshold for surgical technical requirements. Dental LEDs are a readily available resource that causes minimal damage to cells, making them a convenient option for future research and clinical use.

Our work introduces a new approach to regenerating periodontal tissue using a dental LED light to cure collagen hydrogels containing human periodontal ligament fibroblasts applied to the tooth defect area. This engineered construct has the potential to be used for clinical periodontal regeneration.

## 5. Conclusions

The present study demonstrated that HPLF cells combined with LED-illuminated collagen scaffolds promote periodontal tissue regeneration in an SD rat model of periodontal defects. Subsequent to implantation, epithelial cells can be prevented from growing downward, alveolar bone regeneration can be promoted, and epithelial and connective tissue repair can be observed to increase. These finding suggested that the functional and structural recovery of HPLF cells with collagen scaffolds is superior to that of implanted pure collagen scaffolds.

## Figures and Tables

**Figure 1 polymers-15-02649-f001:**
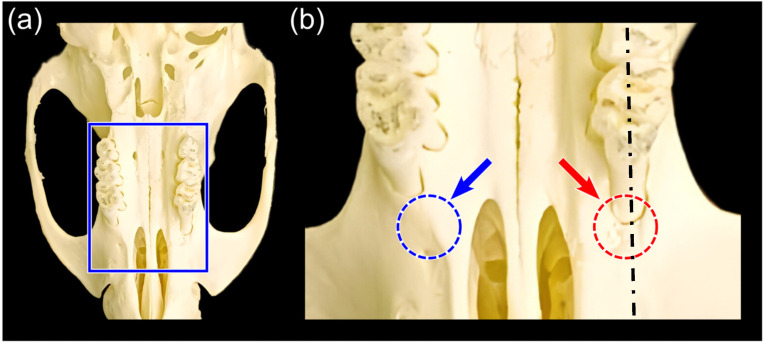
The three-wall defect of the first molar in SD rats. (**a**) The surgical field on the rat skull, indicating the selected sample area that was sectioned. (**b**) Three-wall defect procedures were performed rostral to the upper first molar using a 1.4 mm round tungsten carbide drill. The blue arrow and dotted circle on the left indicate intact bone, while the red arrow and dotted circle on the right indicate the postoperative bone defect, measuring 2 × 1.4 × 1.4 mm³. Note that the dash-dotted line in (**b**) indicates the histological section.

**Figure 2 polymers-15-02649-f002:**
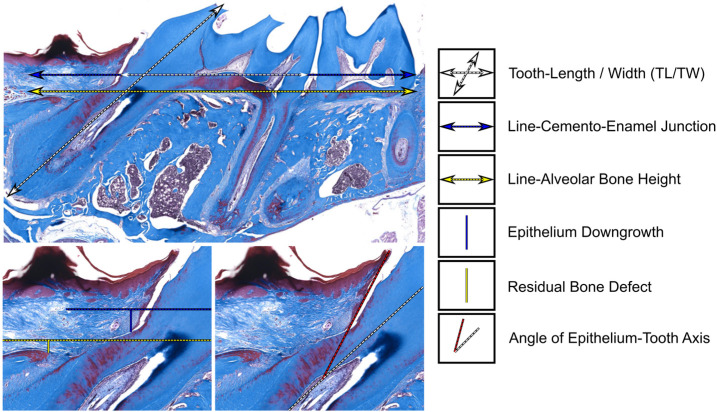
Anatomical reference and calculation method. The cementum–enamel junction and bone height were chosen as anatomical references for the assessment of epithelial downgrowth and residual bone defects, and tooth width and tooth height were also considered to correct for deviations in sample section angles. In addition, the angle of the epi-tooth axis was also evaluated.

**Figure 3 polymers-15-02649-f003:**
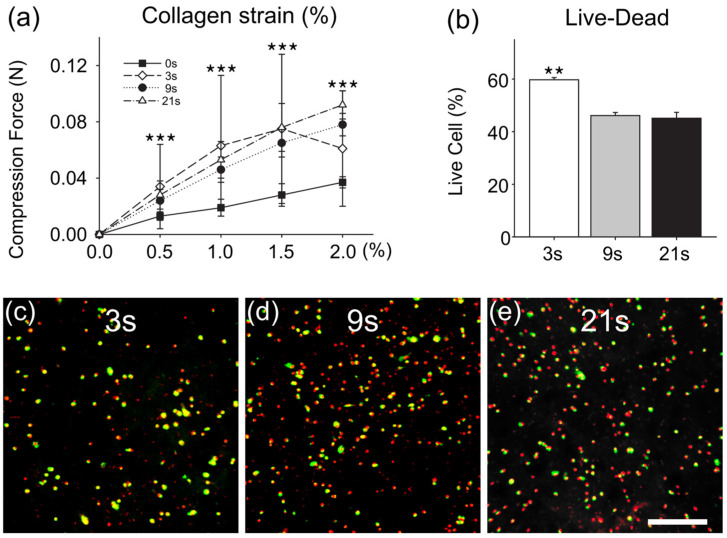
HPLF-laden collagen scaffolds constructed from dental LEDs. (**a**) Strain test (%) of collagen structures cured with the dental LED at different times. The stiffness of the 3 s/9 s/21 s group was higher than that of the 0 s group (*** *p* < 0.001). (**b**) The cell viability under the LED light for 3 s was higher than that of the 9 s and 21 s groups (** *p* < 0.01). (**c**–**e**) Collagen cell scaffolds were illuminated with the dental LED for 3 s/9 s/21 s and cell viability was tested by Live-Dead (bar: 100 µm).

**Figure 4 polymers-15-02649-f004:**
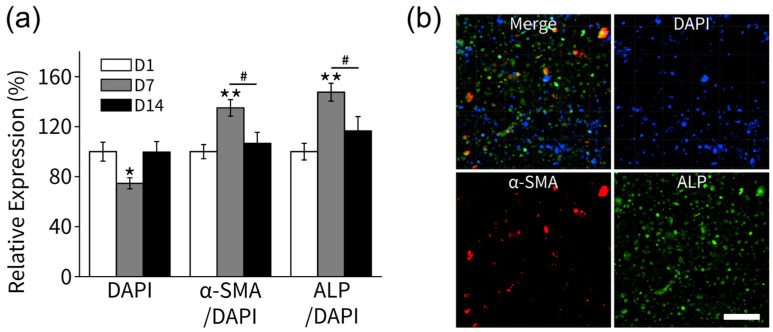
Immunofluorescence staining of α-SMA and ALP in the collagen scaffolds laden with HPLFs (n = 2 for each 1, 7, 14 day experiment). (**a**) Relative fluorescence intensity of DAPI, α-SMA/DAPI, and ALP/DAPI at 1, 7, 14 days (n = 6). (**b**) On the 7th day, both α-SMA and ALP expression were observed, indicating the differentiation of HPLF cells into myofibroblasts and pre-fibroblasts (bar: 100 µm). (* *p* < 0.05 and ** *p* < 0.01, compared with D1; # *p* < 0.05 significant between groups).

**Figure 5 polymers-15-02649-f005:**
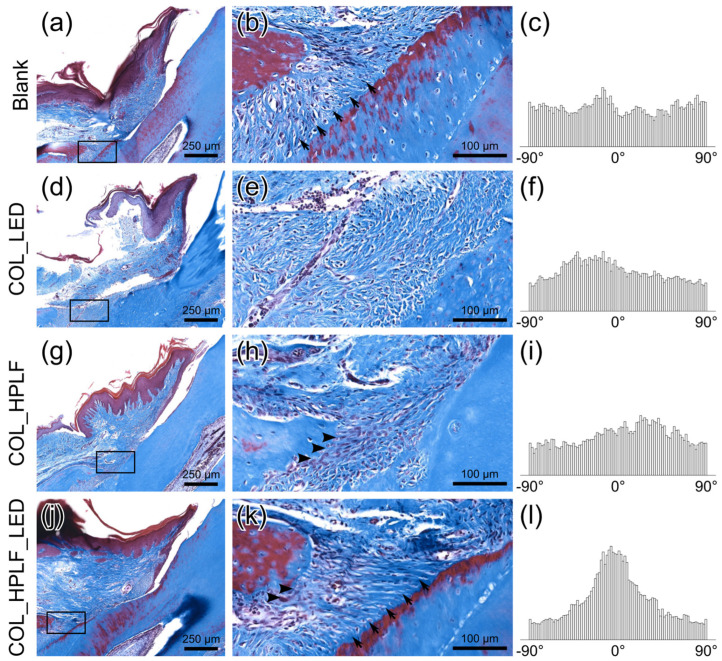
Histology of epithelium healing and bone growth are shown. Movement of the junctional epithelium is a key factor in the progression of periodontitis. The smaller residual bone defect and well-oriented periodontal ligaments indicated better postoperative periodontal regeneration capability. (**a**–**c**) Blank, (**d**–**f**) COL_LED, (**g**–**i**) COL_HPLF, (**j**–**l**) COL_HPLF_LED. (**b**,**e**,**h**,**k**) are enlarged pictures from the small box in (**a**,**d**,**g**,**j**), respectively. The 0° direction represents the orientation of the periodontal ligament fibers that are parallel to the cemento-enamel junction (CEJ) of the tooth. A higher peak at 0° indicates well-organized periodontal ligament fibers. In the COL_HPLF_LED group, osteoblasts aggregated near the newly formed bone, and the periodontal ligament was oriented in connection with the bone and root surfaces. This finding indicated that the periodontal ligament was effectively restored and functioned well in the COL_HPLF_LED group. (Arrows indicate periodontal ligaments, and arrowheads indicate osteoblasts accumulation.)

**Figure 6 polymers-15-02649-f006:**
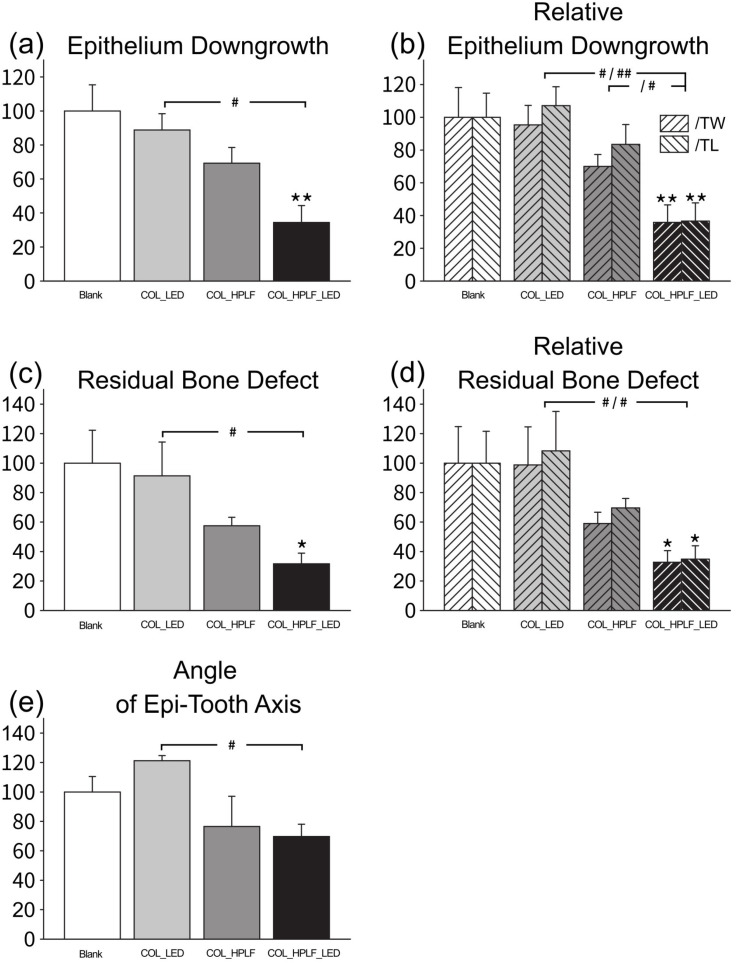
Combination of HPLF cells with LED-illuminated collagen scaffolds promotes periodontal tissue regeneration (Y-axis unit: %). Both (**a**) epithelial downgrowth and (**b**) relative epithelial downgrowth were significantly reduced in the COL_HPLF_LED group compared to the Blank group (** *p* < 0.01) and the COL_LED group (# *p* < 0.05, ## *p* < 0.01). Similarly, (**c**) residual bone defect and (**d**) relative residual bone defect were significantly reduced in the COL_HPLF_LED group compared to the Blank group (* *p* < 0.05) and the COL_LED group (# *p* < 0.05). (**e**) Compared with the COL_LED group, the angle between the epithelium and the tooth axis was significantly reduced in the COL_HPLF_LED group (# *p* < 0.05).

## Data Availability

Data are available upon request.

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
