# Peer review of "Collagen Scaffolds Laden with Human Periodontal Ligament Fibroblasts Promote Periodontal Regeneration in SD Rat Model"

_polymers, 2023, doi:10.3390/polym15122649_

Round 1

Reviewer 1 Report

Thank you for this interesting paper focusing on periodontal regeneration. I would like to congratulate the authors for preparing the present study.

The manuscript is well prepared. However, some corrections are needed. Minor linguistic corrections are necessary.

The title is perfect and precise.

 The abstract seems incomplete. The assessed outcomes and statistical analysis are missing. Please give a short sentence focusing on the clinical impact of the presented results.

 The keywords should be placed in alphabetic order. Please avoid abbreviations (PDL,….)

Introduction: Please give a clear aim of the present study and include a hypothesis and null-hypothesis. Actually, the aim is difficult to understand.

 Materials and Methods: Please give information about an ethical approval of this animal study.

  In the discussion section some points are missing: the study strength, limitations and clinical considerations. Please debate your results regarding the impact on clinical periodontology in our patients. Please give more information about the possible impact of your animal results on human regenerative procedures.

Minor linguistic corrections are necessary.

Author Response

We really appreciate the reviewer's comments and corrections. And we have responded point by point according to the comments as below. Thank you for your scientific insight to increase the depth of our research and also your kindly review helping us achieve better publications.

Point 1:

The abstract seems incomplete. The assessed outcomes and statistical analysis are missing. Please give a short sentence focusing on the clinical impact of the presented results.

The keywords should be placed in alphabetic order. Please avoid abbreviations (PDL,….)

Response 1:

  • Thanks for the comment. We have re-organize the abstract, added important results with statistical analysis, as well as added sentence focusing on the clinical impact of the presented results “The approach developed in this study demonstrates clinical feasibility and holds promise for achieving both functional and structural regeneration of periodontal defects.”
  • Thanks for the comment. We have made the corrections in accordance.

Point 2:

Introduction: Please give a clear aim of the present study and include a hypothesis and null-hypothesis. Actually, the aim is difficult to understand.

Response 2:

  • Thanks for the comment. We have add a sentence to descript the null-hypothesis of this study in Line 100 “The null-hypothesis of this study is that collagen with appropriated mechanical strength combining cell therapy will not affect the PDL orientation, epithelium downgrowth, and bone regeneration of the artificial periodontal defects.”

 Point 3:  Materials and Methods: Please give information about an ethical approval of this animal study.

Response 3:

  • Thanks for the comment. We have mentioned in “Materials and Methods-Line 164 as Animal Care and Use Committee of China Medical University (CMUIACUC-2020-125). The copy was attached below for reference. (only presented in the word file) 

Point 4:   In the discussion section some points are missing: the study strength, limitations and clinical considerations. Please debate your results regarding the impact on clinical periodontology in our patients. Please give more information about the possible impact of your animal results on human regenerative procedures.

Response 4:

  • Thanks for the comment. We have add sentences to focus our study strength, statement the limitations, and clinical considerations as following.

in section 4.6. “Our work introduces a new approach to regenerate periodontal tissue using dental LED light to cure collagen hydrogel containing human periodontal ligament fibroblasts applied to the tooth defect area. This engineered construct has the potential to be used for clinical periodontal regeneration.”

in section 4.5. “This animal model is a surgical defect model, not a disease-induced defect model, so there may be differences in localized tissue phenotype. Disease-induced defect models can cause related diseases due to bacterial infection and inflammation that are difficult to control, and there are individual differences in disease resistance, which can easily cause differences. Surgical defect models are commonly used experimental models because the size of the experimental area is relatively easy to control (Oortgiesen, Plachokova et al. 2012, Yu, Oortgiesen et al. 2013, Cai, Yang et al. 2015)”

in section 4.2. “The Cell-Collagen hydrogel was injected into the surgical defect area using a labora-tory pipette and was sutured after being cured with a dental LED light, which is in line with the clinical operation protocol technique. The COL_LED group formed a barrier like the clinical Guided Tissue Regeneration (GTR), preventing epithelial downgrowth and providing space for bone growth. However, without HPLF cell regulation, there was no significant difference compared to the blank group. Although the COL_HPLF group had HPLF cell regulation, collagen protein toughness was significantly insufficient without dental LED light cross-linking. HPLF cells significantly aggregated at the bottom of the surgical defect area and failed to spread over the entire surgical defect area, so there was no significant difference compared to the blank group. The COL_HPLF_LED group had HPLF cell regulation and dental LED light cross-linking, resulting in significantly improved collagen toughness. HPLF cells were significantly dispersed in the surgical defect area, making bone residual defects or the degree of epithelial migration the lowest. The periodontal tissue regeneration was optimal and the results showed a statistically significant difference.”

Reviewer 2 Report

Review on Chang et al., “Collagen Scaffold ladened with Human Periodontal Ligament Fibroblasts Promote Periodontal Regeneration in SD Rat Model

1.       In the introduction chapter, please, emphasize the importance of the topic with epidemiological data (e.g.: prevalence tendencies of periodontal disease different subtypes at Taiwan, and in the world)

2.       Periodontal ligament fibroblasts =HPLF: do you really need the “H”. What does it stand for?

3.       Please, state the aim of the study in a clearer manner. What is the aim? How it is done, must go to “Materials and methods” section.

4.       In materials and methods 2.6. Animals and surgical protocol section only animals are mentioned. Please, state the number and type of the animals used, at the beginning (rats) not just at the end of the chapter.

5.       2.7 is missing.

6.       I would not say irradiation for blue light. Probably, it is more illumination.

7.       At some sentences the verbs are not in past tense. (e.g.: sacrifice, line 165)

Author Response

We really appreciate the reviewer's comments and corrections. And we have responded point by point according to the comments as below. Thank you for your scientific insight to increase the depth of our research and also your kindly review helping us achieve better publications.

Reviewer 3 Report

The paper studies the impact of a crosslinked and cell-laden collagen hydrogel on the regeneration of peridontal tissue in rat. The authors performed some mechanical testings to show the impact of a riboflavin crosslinking on the mechanical properties of the hydrogel and examined cell viability after crosslinking the cell-laden gel with dental LED. Human periodontal ligament fibroblasts were able to differentiate in vitro into myofibroblasts and osteoblasts in the presence of differentiation media indicating the regenerative potential of such cells in the collagen hydrogel. Finally, the authors analyzed the regeneration after injecting a cell-laden, crosslinked collagen hydrogel in a peridontal defect in SD rats. The study claimes that the peridontal ligaments are restored after injecting the cell-laden, crosslinked hydrogel compared to untreated defects or hydrogels without cells or non-crosslinked cell-laden hydrogels.

General comments:

Introduction

1. The introduction claims that the mechanical strength of a collagen gel is too low for clinical applications. However, the required strength for an application in peridontal regeneration is not clear to me. Since the crosslinking methods is an important piont of the study it would be helpful to add a short summary about what is the current knowledge on the requirements regarding biomaterial strength in peridonatal application.

2. It is further not clear what are the main reasons why the authors have chosen human periodontal ligament fibroblasts as a source of cells.

3. The study analyses the differentiation of the cells within the hydrogel and in the presence of differentiation media. What is the intention of those experiments. I interpreted it as showing the regenerative potential of the cells, but isn't that already known?

For the reader it would be helpful if the authors briefly mention, why the osteoblast and myofibroblast phenotype is important during regeneration in the peridontal ligament.

4. Line 86 describes that the aim of the study is to develop a novel technique to regenerate to well-oriented collagen fibers. There is no description why the orientation of the collagen is important.

Materials and Methods

5. I wonder how can you inject a crosslinked collagen hydrogel into the defect in rats? Crosslinked collagen is not liquid anymore.

6. How often did the authors perform the experiments. It is not mentioned.

Results

7. The results are missing some controls. The in vitro cell viability has been checked for crosslinked hydrogels, but a non-crosslinked gel is missing. Since the authors injected this control into the rats, it would be important to show how the cells behave in the non-crosslinked gel.

8. Why did the authors chose strain rates between 0.5 and 2%?

9. Figure 1: Cell viability is not very good. Do they have an explanation why it is taht low. In the introduction the study mentions that riboflavin is cell compatible. The scale bar has no size indication.

10. Figure 2a): It is not clear what has been used as the 100% control. Cells at day 0, cells in culture dishes ...? In the text it is mentioned: day 1 = 100%, but I can't see this in the graph.

The axis labeling "relative expression" is not very common for analyses on microscopic images. It is more often used for Real Time-PCR results. It is irritating for the reader.

The authors explain the variation in signal on day 7 and 14 with a possible reduction in the proliferation rate during osteoblast differentiation (line 402f)? But Dapi signal goes up from day 7 to 14. Please explain.

How often did the athors see this change in the SMA and ALP signal (in independent experiments)?

11. Figure 2b): Are these images representative images. The SMA signal is very low. I can't see 80% cells expressing SMA.

The cells look in general more roundish. I would expect a different cell morphology for fibroblasts in a collagen hydrogel. Is the viscosity of the gel too low?

12. Figure 4-6: The photos that are shown in all three figures are from one and the same individual for each group. If 24 rats have been tested in three independent experiments, I would expect that more representative images are available for publication. It appears that the authors have expended minimal effort and have performed only 1 in vivo experiment with which to publish quickly. This is not enough.

Non of the image has a scale bar.

13. Figure 4: An overview image would be nice to see, where the image is located within the jaw of the rat. I completely miss a healthy control to compare the regeneration potential. The authors should include it.

From the single picture that is shown for each group it can be concluded that a blank defect is better than injecting a biomaterial. This is contradictory to the literature. How do the authors explain this observation.

Orientation of the collagen fibres must be quantified in order to conclude that one group is better that the other.

The authors marked osteoblasts in the images. It is not clear if the cells come form the rat or from the fibroblasts they injected. Also here I would suggest to count the osteoblasts for comparision.

14. Figure 5: It shows the same animal like in Figure 4d and 6d. Please show different images. And again: a image of a healthy control is missing.

15. Figure 7: Axis in %?

Although the authors included figure 5 to explain the measurements, I still don't understand it. How do the authors know, where epithelium downgrowth starts. What defines the original epithelium heigth? Similar question comes up for the residual bone defect. Is the neighboring bone of standard heigth?

Discussion

16. Line 333 to 336 is not investigated in this study and should be removed. It is not clear if the cells are actively regenerating the tissue or if they die and other mechanisms are activated that induced the migration of cells from the rat into the defect. It would be nice if the others could provide a prove that the human cells are still present in the defect at day 14.

17. Line 351-353: The sentence should be used with care. The bad cellular behaviour after riboflavin crosslinking in vitro could be an indication that the crosslinking is a problem. A control with only collagen and no crosslinking is missing. Similar comment for Line 357 - 359.

18. Line 360 ff: There is no evidence for this. Please remove the sentences or provide measurements of collagen orientation and human cell presence.

Minor changes:

19. Line 69: Sentence is repeated in line 71f.

20. Description Figure 2: Is the first sentence correct "... in ALP and SMA"

21. Line 234: Positon of "Figure 2A" is wrong. Should be within the sentence before.

Author Response

(The authors gave the same response as above.)

Round 2

Reviewer 1 Report

Thank you for adressing all points. The paper is improved significantly.

Minor english correction are necessary and could be done within the final reading.